# Potential Effects of Awn Length Variation on Seed Yield and Components, Seed Dispersal and Germination Performance in Siberian Wildrye (*Elymus sibiricus* L.)

**DOI:** 10.3390/plants8120561

**Published:** 2019-12-01

**Authors:** Fabrice Ntakirutimana, Bowen Xiao, Wengang Xie, Junchao Zhang, Zongyu Zhang, Na Wang, Jiajun Yan

**Affiliations:** 1State Key Laboratory of Grassland Agro-Ecosystems, Key Laboratory of Grassland Livestock Industry Innovation, Ministry of Agriculture, College of Pastoral Agriculture Science and Technology, Lanzhou University, Lanzhou 730020, China; ntakirutimana16@lzu.edu.cn (F.N.); xiaobw15@lzu.edu.cn (B.X.); zhangjch16@lzu.edu.cn (J.Z.); zhangzy@lzu.edu.cn (Z.Z.); wangn17@lzu.edu.cn (N.W.); 2Sichuan Academy of Grassland Science, Chengdu 611731, China; zhifengcao121@163.com

**Keywords:** awns, seed yield components, seed yield, seed germination, seed dispersal, grasses

## Abstract

Awns, needle-like structures formed on the distal of the lemmas in the florets, are of interest because of their essential roles in seed dispersal, germination and photosynthesis. Previous research has reported the potential benefits of awns in major cereal grasses, yet reports on the agronomic and economic implications of awn length variation in forage grasses remain scarce. This study investigated the variation of awn length among 20 Siberian wildrye populations and the effect of awn length on seed yield and yield components. This work then studied the impact of awn length on seed dispersal and germination. The analyses indicated a high level of awn length variation among populations. Awn length showed a significant influence on harvested seed yield per plant (*p* < 0.05) mostly driven by interactions between awn length and the majority of seed yield components. Principal component analysis clearly revealed that the final impact of awn length on seed yield depends on the balance of its positive and negative effects on traits determining seed yield. Furthermore, awn length tended to increase seed dispersal distance, although little diversity in the nature of this progression was observed in some populations. Awn length exhibited a significant relationship (*p* < 0.05) with germination percentage. It also tended to shorten germination duration, although this interaction was not statistically significant. Collectively, these results provide vital information for breeding and agronomic programs aiming to maintain yield in grasses. This is the first report to demonstrate in Siberian wildrye the agronomic impacts of awn length variation.

## 1. Introduction

Siberian Wildrye (*Elymus sibiricus* L.), which belongs to the genus *Elymus* (Poaceae: Triticeae), is an important perennial, self-pollinating and cool-season bunchgrass. The grass is native to the high altitude regions of western and northern China [1] and widespread in Asia, Europe and North America [2]. This species has erect naked stems with evenly distributed basal leaves, providing good grazing for livestock. The inflorescence of Siberian wildrye is a long, flexuous spike containing about 30 spikelets, in which each spikelet consists of 5–8 florets with outcurving long awns that can repel seed-eating birds and mammals [3]. Unlike domesticated cereal crops, such as rice and wheat, that have awnless species, which facilitate harvesting and processing activities [4], no awnless Siberian wildrye genotype has been reported so far. Siberian wildrye has been widely used in cultivated pastures and natural grasslands because of its high quality and palatable forage with high protein content, as well as excellent stress tolerance [5]. However, the grass is biologically ineffective at producing seeds and there are considerable seed losses even in suitable growing conditions [2]. Fluctuations in seed development and subsequent yield in Siberian wildrye, as in many other grass species, may be influenced by growing conditions, physiological processes underlying assimilates production and partitioning and molecular genetic factors [6,7]. Consequently, forage breeders have mainly focused on increasing biomass yield and enhancing forage quality [8]. Despite this, seed yield remains a decisive factor for a cultivar success in this species and understanding traits determining seed yield potential is imperative [2].

Seed yield in grasses is known to be associated with a number of seed yield components, such as spikelets per inflorescence, seeds per inflorescence, seed size and seed weight and seems impossible to achieve realized seed yields without considering these traits [6,9,10]. A number of considerable works have shown awns are positively associated with larger seed size and thereby increased yield in less favorable conditions [11,12]. Under favorable growing conditions, however, long awns compete with florets for assimilates, which exacerbates assimilate partitioning to florets. As a result, an increase in the number of sterile spikelets and thereby a decrease in harvested seed yield is detected [13]. Hence, the actual benefits of awns on seed yield remain controversial due to the lack of clear relationships between awn characteristics and traits determining seed yield. Studies investigating the photosynthetic role of awns compared to that of culm (mainly flag leaves) have applied several methodologies, such as source-sink manipulation (shading or/and removal of awns), the application of heavy radioactive isotope ^14^C and stable radioactive isotope ^13^C and carbon isotope discrimination to assign the contribution of awns to grain filling and harvestable yield but the obtained results varied greatly [14]. This variation seems to be associated with applied protocols and genotype x environment interaction [15] but biases related to sampling methods and phenotyping tools should be taken into account. Perhaps the most feasible approach to understanding the benefits of awns is to evaluate awn length variation and its direct and indirect effects on traits determining seed yield. As awn length is not only variable among species but also within species, individuals of the same population and even among spikelets of the same inflorescence [16,17], precise phenotyping tools from species representing a range of diverse genetic backgrounds could contribute toward understanding the role of awns on yield potential. Awn length variation provides a foundation for the selection and improvement of components of seed yield thereby realizing high seed yields.

Seed dispersal and germination are critical traits with the potential to evaluate the economic and ecological fitness of plant species [18,19]. There is increasing evidence that seed traits are coupled with both seed dispersal and germination performance. It is reported that larger seeds are disadvantageous during dispersal but contain more carbohydrates reserves, which facilitate seed germination and give rise to more vigorous seedlings than smaller ones [20,21]. A number of studies have shown that long and barbed awns assist seed dispersal by adhesion to animal fur [22,23,24]. However, reports on the contribution of awns on germination behavior remain scarce. A previous study indicated long awns could help seeds to rapidly emerge from deeper soil depth [25] but this is also impacted by several factors, including soil characteristics and coleoptile length, a concern appears to reveal the elusive role of awns on seedling emergence [26]. Under natural conditions, awned seeds’ burial depth during dispersal is deeper than that of awnless seeds, promoting initial seed growth especially in high temperature and fire-prone environments [16]. Therefore, seed dispersal and germination as critical stages of the life cycle of plant species could aid to assess the potential ecological and economic implications of awn length in grasses.

In contrast to major cereal crops, the agronomic and economic implications of awns in forage grasses are still poorly understood and specific information on the benefits of awns in Siberian wildrye is lacking. The aims of this study were to evaluate the variation of awn length across 20 Siberian wildrye populations and assess the influence of awn length on seed yield and seed yield components such as spikelets per inflorescence, florets per spikelet, seed set (%), seeds per inflorescence, seed size and thousand-seed weight (TSW). We then studied the impact of awn length variation on seed dispersal and germination performance. Our results provide fundamental information for the selection and future genetic improvement of awn traits in this species.

## 2. Results

### 2.1. Variability in Awn Length, Yield Componentsand Seed Yield 

The morphological variation for seed traits among 20 Siberian wildrye populations was investigated. Four of nine traits—awn length, seed set (%), seeds per inflorescence and TSW exhibited the coefficient of variation (CV) value higher than 10% (Table 1). The highest morphological variation was found for awn length (CV = 22.26%), while the lowest was found for florets per spikelet (CV = 2.55%). Seed yield per plant exhibited a moderate variation among 20 Siberian wildrye populations with a CV of 6.43%.

Clustering of nine traits formed two groups (Figure 1). The first group included five traits (awn length, seed length, seed width, TSW and seed yield per plant). The remaining four traits were assigned into the second group. Clustering analysis revealed that populations with the same geographical origin and similar phenotypic traits tended to group together. A total of 18 populations were assigned into group one and two populations, ZN02 and ZN03, were assigned in group two. The dendrogram was consistent with variation in seed traits. 

To assess the effects of population and spikelet position on awn length, seed yield per plant and yield components, the spike was divided into three equal parts (basal, central and apical) based upon the spike length. Results of the ANOVA revealed differences between populations for awn length and seed width at particular spikelet positions (Table 2). There were significant differences among Siberian wildrye populations for awn length, seed yield per plant and all measured seed yield components (*p* < 0.05), except for florets per spikelet (Table 3). Awn length across populations ranged from 6.30 mm (LQ06) to 19.39 mm (ZN02), with an average of 12.41 mm (Table 1 and Table 3). Three populations (LQ06, XH04 and TZ02) showed a mean awn length of less than 10 mm (Figure 2). 

Seed lengths ranged from 8.37 mm (LQ01) to 10.48 mm (TZ01) (Table 3). The majority of seed lengths (about 75%) were below 10 mm, with the seeds of ZN02 significantly longer than that of LQ01, LQ03, XH04, LQ06 and HZ01. Whereas LQ01 had narrower seeds than any other population tested here, ZN02 showed wider seeds than any other population. Spikelets per inflorescence ranged from 19.34 (ZN02) to 27.77 (ZN01), with minor differences in the number of spikelets observed between short-awned and long-awned populations. Florets per spikelet ranged from 5.80 (HZ01) to 6.47 (XH01), with a narrow difference observed among populations. Analyses across populations showed great variation in seed set (%), ranging from 38.50 (ZN02) to 73.09 (LQ06). Among populations, the highest and lowest frequencies of seeds per inflorescence were 55.25 observed in ZNO2 and 93.86 observed in HZ01, respectively. The mean weight per 1000 seeds was 3.89 g. The heaviest and lightest seeds were found in ZN02 (4.7 g) and LQ06 (3.05 g), respectively. Seed yield per plant ranged from 7.60g (LQ06) to 9.99g (ZN03), with a mean value of 8.76 g (Table 1 and Table 3). 

### 2.2. Relationships Between Awn Length, Yield Componentsand Seed Yield 

Correlation analyses indicated a wide range of relationships between awn length and seed yield components and among seed yield components (Table 4). Analyses revealed a significant correlation between awn length and seed length (r = 0.474). Seed set (%) and seeds per inflorescence showed a significant negative relationship with awn length with Pearson correlation coefficients of −0.677 and −0.722, respectively. Seed length and seed width showed a significant positive relationship with TSW with r values of 0.466 and 0.604, respectively. There was a significant relationship between seed set (%) and seeds per inflorescence (r = 0.519). Seed length showed a significant positive correlation with seed width (r = 0.428). However, a significant negative relationship was found between seed length and seed set (%) and seeds per inflorescence with Pearson correlation coefficients of −0.369 and −0.447, respectively (Table 4). 

Analyses among populations revealed the contributions of some traits to the final seed yield per plant. Pearson correlation analysis revealed a significant positive relationship between awn length and seed yield per plant (r = 0.544) (Table 4). Spikelets per inflorescence exhibited a significant positive correlation with seed yield (r = 0.371). Whereas seed set (%) showed a significant negative relationship with seed yield (r = −0.231), florets per spikelet and seeds per inflorescence tended to decrease seed yield per plant, although these relationships were not statistically significant. Seed yield was significantly correlated with seed length (r = 0.487) and seed width (r = 0.448). In addition, a significant positive correlation between TSW and seed yield per plant (r = 0.526) revealed an overall positive seed weight- seed yield relationship.

The first three components of the principal component analysis (PCA) showed eigenvalues greater than 1 and explained 72.65% of total phenotypic variation (Table 5). The first component (PC1) had large positive associations with awn length (0.886), TSW (0.832) and seed length (0.706). PC1 also showed larger negative associations with seed number and seed set (%), with coefficients of −0.702 and −0.696, respectively, suggesting that PC1 was mainly involved in the impact of awn length on seed yield and components. While the second component (PC2) mainly revealed the contribution of yield components to seed yield per plant, it showed large positive associations with spikelet per inflorescence (0.773), seed yield per plant (0.593) and seeds per inflorescence (0.491). The third component (PC3) had large positive associations with florets per spikelet (0.841) and spikelets per inflorescence (0.399). It showed higher negative associations with seed set (%) with a coefficient of −0.418. PC3 focused mainly on relationships among seed yield components.

### 2.3. Impact of Geographical Distribution on Variation of Awn Length, Seed Yield Components and Seed Yield

Pearson correlation analysis was conducted to test the effects of environmental factors such as latitude, longitude and altitude on variation in awn length, seed yield components and seed yield. Analyses across 20 Siberian wildrye populations showed that awn length exhibited a negative correlation with latitude (r = −0.189) and longitude (r = −0.400) but it showed a weak positive correlation with altitude (r = 0.094). Seed length, florets per spikelet, seed set (%) and seeds per inflorescence exhibited a positive correlation with latitude, though no one was significant (Table 4). Seed width, spikelets per inflorescence and TSW showed a negative relationship with latitude. All seed traits, except seed set (%) and seeds per inflorescence, exhibited a negative correlation with longitude with correlation coefficients ranging from −0.127 for spikelets per inflorescence to −0.663 for seed length. Seed width, spikelets per inflorescence and TSW showed a positive, though non-significant, relationship with altitude. In addition, seed length, florets per spikelet, seed set (%) and seed per inflorescence exhibited a negative relationship with altitude, also non-significant (Table 4). Thus, geographical factors (latitude, longitude and altitude) generally showed a negative correlation with seed yield per plant, while seed yield per plant decreased significantly with increasing longitude.

### 2.4. Pattern of Seed Dispersal Distance and Germination Traits

Seed dispersal distance was measured under a specific wind speed and showed variation across 20 Siberian wildrye populations (Table 6). ANOVA test revealed significant differences among populations for seed dispersal distance (*p* < 0.05). Seed dispersal distance increased with increasing awn length, although little diversity in the nature of this progression was observed in some populations. The longest and shortest seed dispersal distances were 1.68 m observed for HZ01 and 1.02 m observed for ZN02, respectively.

Under controlled germination experiments, germination traits were measured across Siberian wildrye populations. Results of the ANOVA showed significant differences among populations for germination indices measured here (*p* < 0.05), except for germination duration. The highest and the lowest germination percentages were 78.00% found for ZNO2 and 97.00% observed for LQ06, respectively (Table 6). Germination index ranged from 2.21 (HZ01) and 3.10 (TZ01), with significant differences documented among several populations. The longest and shortest duration for the completion of the germination process were 5.20 days observed in ZN01 and 3.80 days observed in XH01 and TZ03, respectively. Germination delay ranged from 3 days and 3.80 days and did not differ significantly among many populations. 

### 2.5. Effects of Seed Traits on Seed Dispersal Distance and Germination Behavior

Analyses indicated that awn length tended to increase seed dispersal distance although this interaction was not statistically significant (r = 0.059) (Table 7). All measured traits showed a positive relationship with seed dispersal distance with a Pearson correlation coefficient ranging from 0.034 for seed length to 0.128 for seed width. The analysis showed that germination percentage significantly decreased with increasing awn length with a Pearson correlation coefficient of −0.357 (Table 7). Seed length and seed width exhibited negative relationships with germination percent, indicating that the germination rate decreased with increasing seed size. The TSW showed a significant negative correlation with germination percent (r = −0.241), indicating that populations with heavier seeds exhibited a low germination rate. Seed germination delayed with increasing seed width and TSW with Pearson correlation coefficients of 0.139 and 0.031, respectively. 

The PCA among seed traits, germination indices and seed dispersal distance indicated that the first three components showed eigenvalues greater than 1 and explained 61.714% of total variation (Table 8). The first component (PC1) showed large positive associations with TSW (0.858), awn length (0.803), seed width (0.717). PC1 also showed large negative associations with germination percentage (−0.516) and germination index (−0.276), suggesting that this component was mainly involved in the effect of seed traits (TSW, awn length and seed width) on germination rate. The second component (PC2) showed larger positive associations with germination index (0.821) and seed length (0.396) and showed large negative associations with germination delay (−0.644) and germination delay (−0.605), indicating that longer seeds improved germination rate and shortened germination duration. The third component (PC3) showed positive associations with seed dispersal distance (0.672), seed width (0.327) and TSW (0.110). It also showed negative associations with awn length (−0.132) and seed length (−0.043). PC3 focused mainly on the impact of seed traits have on seed dispersal distance.

## 3. Discussion 

### 3.1. Impact of Awn Length on Seed Yield and Components

A number of yield components, such as spikelets per inflorescence, seeds per inflorescence, seed size and seed weight, have the potential to influence harvested seed yield [27,28,29]. The large influence of awn length on harvested seed yield is due to its interaction with yield determining traits. Notably, our results showed a strong negative correlation between awn length and seeds per inflorescence (Table 4), indicating that seed set (%) and the number of seeds in the florets decreased significantly with increasing awn length. Although awn length increased with a decreasing number of seeds per inflorescence, awn length showed a significant positive correlation with two major seed yield components (seed length and seed width) across populations (Table 4). In addition, PCA results showed a wide variation of the contribution of awn length to seed yield components, indicating both positive and negative associations. Awn length had a great influence on harvested seed yield per plant via indirect causal effects that led to large associations with traits determining potential seed yield (Table 5). This indicates that long-awned plants might contribute to improving seed yield than short-awned plants in Siberian wildrye populations. The significance of awn length variation was previously investigated and was linked with several physiological processes. Otamane Merah et al. [15] used a carbon isotype discrimination approach in wheat and found that ear organs, including awns, contributed more to grain filling than flag leaves. The contribution of awns to kernel filling and grain yield was also supported by Rut Sachez-Bragado et al. [14]. In contrast, awns showed no significant or lower contribution to grain filling and grain yield increment compared to flag leaves when awn removal experiments were performed [30]. These results indicate that the contribution of awns on seed yield depends not only on its direct and indirect interactions with seed yield components but also on morphological features of awns.

While floret fertility is mainly associated with assimilate partitioning to the spikelets [12,28], increasing the number of fertile florets has been believed as a potential alternative to increase yield in grasses [13]. In the current study, floret fertility decreased with increasing awn length to reduce seeds per inflorescence (Table 4 and Table 5). Although long awns reduced seed number in Siberian wildrye, this trade-off showed a minor impact on seed yield per plant. Instead, correlation analyses indicated that seeds per inflorescence exhibited a total negative contribution on seed yield (Table 4). Previously, seed number varied significantly and showed a slight contribution to harvested seed yield in *Leymus chinensis* [10]. Other studies obtained similar results in wheat [9,12] and *Stipa pupurea* [31]. Contrarily, in some species, such as rice, reduced grain number by awns decreased harvested grain yield, despite their contribution to grain size increment [32], implying that seed yield increment by long awns depends on a significant increase in seed size that compensates reductions in seed number.

TSW increased with increasing seed size, indicating its great impact on seed yield. Supporting this observation, TSW exhibited a strong association with seed length and seed width when PCA was conducted, suggesting a positive correlated response in these traits which influenced seed yield (Table 5). These results reflect that seed size and seed weight were the most important seed yield components in Siberian wildrye populations. Similarly, the experiments by Hebblethwaite and Ivins [33] and Abel et al. [6] stated TSW contributed to greater harvestable seed yield in ryegrass (*Lolium perenne* L.). In wheat, awns increased grain size and thereby harvested grain yield [9,34]. As indicated, awns interacted with the majority of measured seed yield components. It is noteworthy that although the increment of assimilates by awns may contribute to floret fertility, the fact that long awns may compete with growing kernels and thereby reduce floret survival, especially under good agronomic conditions, is another consideration [9,12,34]. Thus, seed yield improvement via awn length optimization should be suggested as the breeding strategy. 

Our results also showed an association between spikelets per inflorescence and traits determining seed yield potential such as seed size, seeds per inflorescence and TSW. These findings are consistent with a previous study that spikelet number has a significant direct effect on seed yield [6]. Moreover, evidence in wheat [13] and barley [4] suggested no relationship between awn length and spikelet number. Although the results of correlation analysis of the current study across Siberian wildrye populations supported this argument (Table 4), when PCA was performed, spikelets per inflorescence showed negative associations with awn length, though weak (Table 5). This suggests a likely possible interaction between awn length and spikelet number, which should be targeted by further experimental studies. While fluctuation in seed yield and components is common among and within grass species [35], it is thus worth noting that analyses derived from single yield component, individual plant, single plant population or single plant species may lead to misinterpretation of the impact of awn length on seed yield and components. A balance and interaction between different yield components may be the key factors that regulate and affect seed yield. Furthermore, seed yield is affected by the interaction of seed yield components, genetic mechanisms and agronomic and environmental factors [2,6,31]. Analyzing this interaction in different grass species would be a valid method for identifying awns’ impact on seed yield.

### 3.2. Seed Yield and Components as Affected by Geographical Factors

Geographical factors, such as latitude, longitude and altitude, have been regarded as the major contributors to yield variation in several plant species [36]. It was shown that changes in latitude, longitude and altitude may result in variation in temperature, radiation, moisture and soil fertility, which influence seed yield and components [31,35]. Previous data indicate latitude exhibited a positive correlation with seed traits, whereas longitude and altitude showed a negative correlation with seed yield-related traits in several species [37,38]. The results of the current study differ with this observation, although some seed traits showed a positive correlation with latitude and a negative correlation with longitude and altitude (Table 4). This can result from dramatic changes in seed yield components among Siberian wildrye populations, which accounted for the variation in the interactions among seed yield components and geographical factors. This supported a previous finding that seed yield and components in plant species may differ in adverse conditions across regions [39]. 

In the current study, Siberian wildrye populations varied spatially and temporally, indicating a wide variation observed in the relationships between geographical factors and seed yield and components. Our analyses revealed that seed yield per plant decreased with increasing latitude, longitude and altitude (Table 4), which reflects the negative interactions among seed yield components and geographical factors. Yield components such as spikelets per inflorescence, TSW and awn length showed a negative relationship with latitude. Awn length and seed set (%) exhibited a positive relationship with longitude. Awn length and TSW exhibited a positive correlation with altitude. Interestingly, other study has found negative correlations between seed mass and geographical factors, such as latitude and longitude, in several species [40]. These findings indicated that geographical factors have multi-lateral impacts and complex interactions with seed yield and components. 

It is thus noteworthy that although recent studies attempted to show the relationship between geographical factors and variation in seed yield, studies on the interactions between awn length and geographical factors and their effects on seed yield are scarce. The observation that awn length was negatively correlated with latitude and longitude but positively correlated with altitude (Table 4) reflected that awn length varies with environmental variables across locations. A previous study has obtained a negative correlation between awn length and geographical conditions, such as latitude and altitude [31]. Collectively, the majority of correlations between geographical factors and seed traits observed here were not significant but the observed variation in seed traits across locations indicated that seed yield in Siberian wildrye was impacted by interactions between geographical factors and seed yield components. 

### 3.3. The Impact of Seed Traits on Seed Dispersal Distance 

Seed dispersal is critical for plant establishment and survival, especially in wild conditions. This process helps seeds to reach suitable conditions for their germination and development [22]. Recently, several studies have reported the potential relationship between long awns and seed dispersal, in which long awns triggered seed dispersal and thus propelled the seeds into the ground [16,21]. However, only a few studies evaluated the impact of awn length on seed dispersal distance. Li et al. (2015) [27] reported a weak positive relationship between awn length and seed dispersal distance in *Stipa pupurea*. In the current study, awn length exhibited a positive relationship with seed dispersal distance across Siberian wildrye populations, although this interaction was not statistically different (Table 7). This indicated that awn length interacts with traits affecting seed dispersal in Siberian wildrye. Besides, some studies have demonstrated that environmental factors, such as wind, may affect seed dispersal under the aid of seed appendages, such as awns [23,41], which implies that awns may affect seed dispersal, especially under wild conditions. In our study, seed dispersal distance showed a negative correlation with seed length, indicating that longer seeds are not easily transported by seed dispersal agents. Notably, TSW showed a large positive association with seed dispersal distance (Table 8). This illustrates how larger seed shortens seed dispersal distance, despite being easily shattered. It also explains how the dispersal units of longer and larger seeds are more easily buried into the soil but not easily dispersed. Collectively, these observations suggest that seed dispersal distance may be controlled by several interacting seed traits.

### 3.4. The Impact of Seed Traits on Germination Indices 

Seed germination can be evaluated by many indices, such as germination percentage, germination index, germination duration and germination delay [42]. Testing methods, seed traits, species, genotypes and environmental conditions are important factors influencing seed germination [20]. Seed germination trend is undoubtedly connected with seed traits, such as seed size, rigidity, maturity and presence of seed appendages, such as awns, wings and hairs [33,34]. Johnson and Baruch [16] reported that long awns could improve seed germination and growth, especially under natural growing conditions. Other study indicated the rapid emergence of awned seeds than awnless seeds may be due to their high carbohydrates reserves [25]. A laboratory study reported a negative correlation between awn length and seed germination rate in *Stipa pupurea* [31]. Our analyses showed a significant negative relationship between awn length and germination percentage among Siberian wildrye populations (Table 7). Since seed traits exhibited high variation among populations (Table 2), our study observed adverse interactions between seed traits and germination indices (Table 7). Given studies investigating the role of awns on seed germination have found inconsistent results, awn length-seed germination interaction remains unclear. Thus, future studies should investigate the interaction between awn features and germination traits under field and greenhouse conditions. 

## 4. Material and Methods

### 4.1. Study Area

Seeds of 20 Siberian wildrye populations were collected from 18 different sites in South Gansu province, China (Figure 3). The collection sites were located across from 35.22°N to 36.96°N in latitude, from 102.82°E to 103.15°E in longitude and from 2380 m to 3380 m in altitude and all seed samples were collected between August and September in 2018 based on a population-based sampling method [43]. Geo-coordinates (latitude, longitude and altitude) were recorded using a Global Positioning System (GPS) for each population (Table 9).

### 4.2. Sampling and Measurement

#### 4.2.1. Seed Yield Measurement

In the early of August, when seeds matured, a total of 10 plants of each population were randomly selected and the heads of each plant were carefully hand-cut. Seed samples were taken from the center of each plot to avoid border effects. Because of serious seed shattering reported in this species [1], seeds were harvested when the seed moisture content was approximately 45% [44]. Seeds of each plant were air-dried, carefully threshed, cleaned, weighed and stored in paper bags at 20 °C and 15% relative humidity before subjected to seed yield evaluation and other seed measurements. Seed yield per plant (SYP) measurement and other laboratory analyses were carried out when the seed moisture content was about 10%. 

#### 4.2.2. Evaluation of Awn Length and Seed Yield Components 

A total of 20 random inflorescences of each individual plant of each population were selected to obtain awn length and seed traits measurement. Spikelets per inflorescence, florets per spikelet, seed setting rate and seeds per inflorescence were determined for each plant. Seed set (%) was determined as the percentage of florets that have filled seeds [2]. For awn length, seed length and seed width, seeds were measured using seeds sampled in the basal, central and distal spikelets; thirty random seeds were chosen in each sample. These seeds were scanned by Epson Perfection V700 Scanner (Seiko Epson Corp., Nagano, Japan) and the images were analyzed using Image J software [45]. To obtain thousand-seed weight (TSW), thirty random inflorescences from each plant were selected and carefully threshed. Three thousand-seed samples from basal, central and distal spikelets were counted and weighed. 

### 4.3. Seed Dispersal Distance Measurement 

To measure the seed dispersal distance, we analyzed seed flight distance under stable wind speed generated by an electric fan. A total of 20 random spikes were selected in each individual plant of each population; three samples each composed with thirty dry and filled seeds were randomly selected from basal, central and distal spikelets of each spike. The seeds were identically released at a height of 20 cm above the ground in front of the stable wind speed (4.8 m/s) generated by an electric fan. Prior to the experiment, the fan was tested for stability and accuracy and a wind speed meter was used to determine the speed of the wind generated by the fan. The horizontal distance that the seeds moved until they reached the ground was recorded [31]. 

### 4.4. Seed Germination Test

To investigate the relationship between seed traits and germination performance, we performed germination test in climate control chambers in Lanzhou University, Gansu, China, from 15 October 2018 to 15 November 2018. Seeds were sterilized for 5 min in 1% NaClO (*w*/*v*), washed 5 times in sterile distilled water and then allowed to dry on filter paper. The sterilized seeds were germinated in plastic boxes containing moistened double layer of blotter paper at 25 °C, 85% relative humidity and a photoperiod of 12 h light—12 h dark. The replicates of fifty seeds chosen from basal, central and distal spikelets were grown in each plastic box. To prevent desiccation, distilled water was daily added to the plastic boxes. Seeds were considered germinated when the seminal root was about 1 mm long and visible. The germinated seeds were noted daily for 30 days. The germination percentage, germination index, germination duration and germination delay were estimated as previously described by Li et al. [31].

### 4.5. Statistical Analyses

Statistical analyses of seed yield and its components, germination indices and seed dispersal were performed from the data that were first checked for normality and homogeneity of variances and that were log-normalized when necessary. Data were then subjected to analysis of variance (ANOVA) and the significant differences between means were determined by Tukey’s Honest Significant Difference (HSD) test at *p* < 0.05 level. To analyze the variability of seed yield and components among Siberian wildrye populations, a heatmap was generated with Heatmap Illustrator (HemI, version 1.0.3.7) software. The visualized color matrix was constructed, in which lines and columns represented populations and seed traits, respectively. The average linkage method and the similarity metric with Pearson distance were used to carry out the hierarchical cluster analysis [46]. Principal component analysis (PCA) and correlation analyses were performed. The correlation coefficients (r_xy_) between awn length and seed yield and components, seed traits and seed dispersal and germination were analyzed with Pearson’s regression analysis (n = 200, that is, total observations of each trait which obtained from 10 plants of each of 20 Siberian wildrye populations). Principal component analysis was performed with the first three principal components, accounting a large amount of total variance retained for further interpretation.All statistical analyses were conducted using SPSS 25.0 software (IBM, Somers, NY, USA). 

## 5. Conclusions

Our study uncovered the relationship between awn length and seed yield and components, seed dispersal and seed germination. Notably, our results showed an increase in awn length leads to a decrease in seeds per floret. Despite this drop off in seed number, awn length contributed to increment in seed yield per plant through larger seed size. The practical implication of these findings is that awns should be targeted for enhancing seed yield in grasses. Despite variation in the relationships between seed traits and seed germination indices, our analyses showed a significant negative correlation between awn length and germination percentage. Awn length also tended to decrease germination duration. These observations indicate that awn length may affect germination indices in grasses. Furthermore, our results suggested a possible link between seed traits and seed dispersal distance. Collectively, there are several factors coupled with plant emergence, development and performance and the present data can only provide information on the impact of seed traits variation on seed yield and their role in seed dispersal and germination. Despite certain limitations, these results provide comprehensive insights to breeding and agronomic programs targeting to maintain seed yield and they add useful scientific information for future seed dispersal and germination studies. Future studies should focus on the interactions between awn length, genetics and environment and their potential impacts on population dynamics and seed yields in grasses. 

## Figures and Tables

**Figure 1 plants-08-00561-f001:**
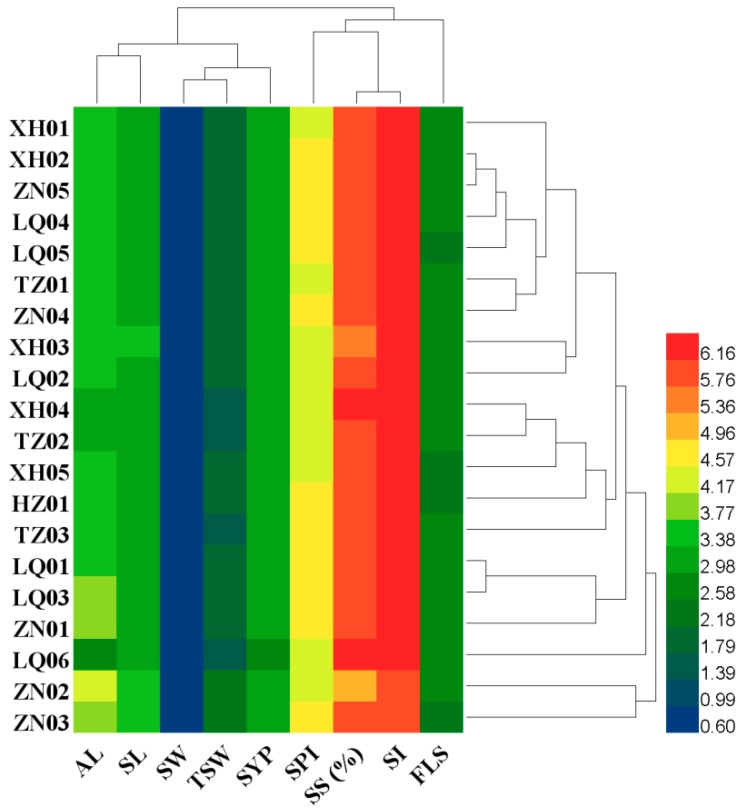
Heatmap and clustering analysis of 20 Siberian wildrye populations using seed traits. Names of populations are represented on the left side of the heatmap. Lines indicate populations, whereas columns represent seed traits. The color scale ranged from 0.57 to 6.16 (green to red). AL, awn length; SL, seed length; SW, seed width; TSW, thousand-seed weight; SYP, seed yield per plant; SPI, spikelets per inflorescence; SS (%), seed set percentage; SI; seeds per inflorescence; FLS, florets per spikelet.

**Figure 2 plants-08-00561-f002:**
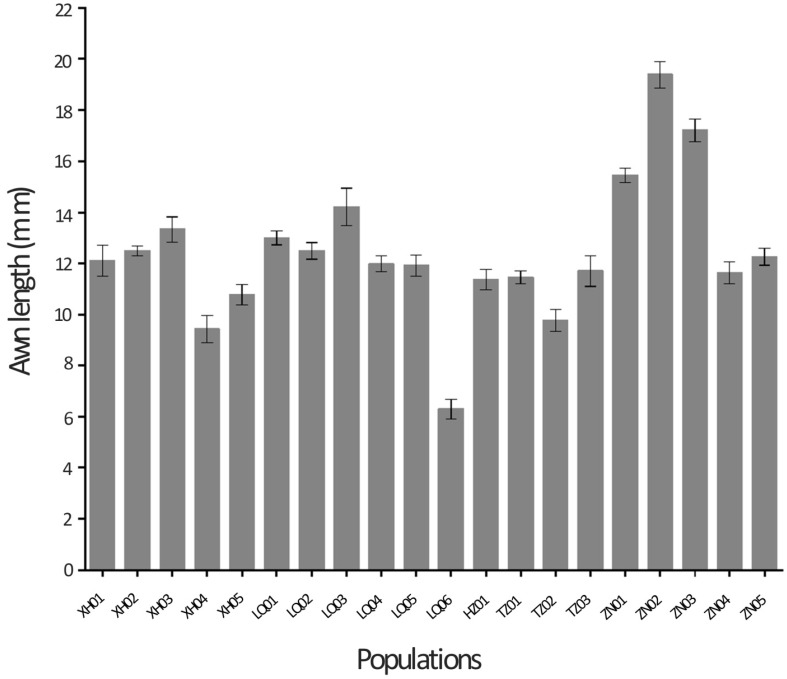
Awn length of 20 Siberian wildrye populations collected in several parts of southern Gansu during 2018. Bars denote the mean values ± standard deviation.

**Figure 3 plants-08-00561-f003:**
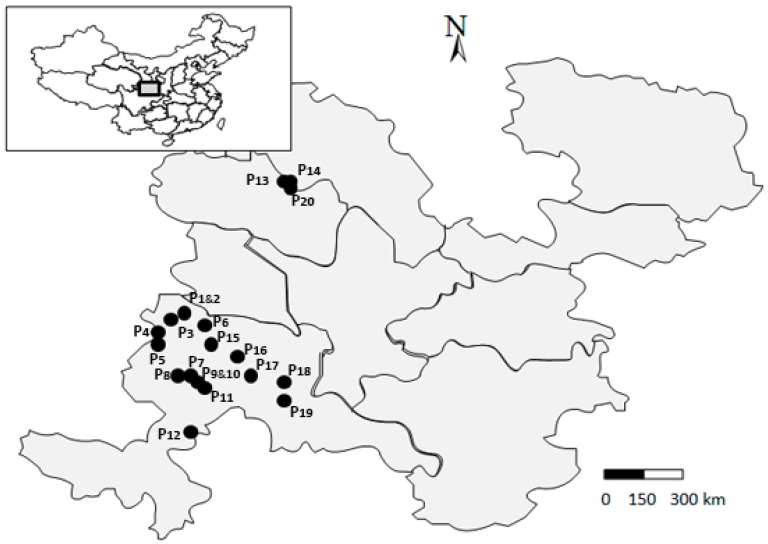
Location of lands of southern Gansu where samples were collected. Dots on the map represent collection sites of Siberian wildrye populations used in the study.

**Table 1 plants-08-00561-t001:** Morphological variation of awn length, seed yield components and seed yield per plant among 20 Siberian wildrye populations.

Traits	Minimum	Maximum	Mean	SD	CV (%)
AL (mm)	6.30	19.39	12.41	2.76	22.26
SL (mm)	8.37	10.48	9.40	0.69	7.32
SW (mm)	1.51	1.76	1.62	0.07	4.42
SPI	19.34	27.77	23.82	2.23	9.35
FLS	5.80	6.47	6.10	0.16	2.55
SS (%)	38.50	73.09	60.76	8.04	13.24
SI	55.25	93.86	86.76	10.41	11.99
TSW (g)	3.05	4.70	3.89	0.43	10.95
SYP (g)	7.60	9.99	8.76	0.56	6.43

AL, awn length; SL, seed length; SW, seed width; SPI, spikelets per inflorescence; FLS, florets per spikelet; SS (%), seed set percentage; SI, seeds per inflorescence; TSW, thousand-seed weight; SD, standard deviation; CV, coefficient of variation.

**Table 2 plants-08-00561-t002:** Variance analysis of awn length, seed yield per plant and yield components as affected by population and spikelet position.

Source of Variance	df	AL	SL	SW	SPI	FLS	SS (%)	SI	TSW	SYP
Population (P)	19	***	***	***	***	ns	***	***	***	***
Spikelet position (Sp)	2	**	ns	*	ns	ns	ns	ns	ns	ns
P × Sp	38	***	ns	**	ns	ns	ns	ns	ns	ns

AL, awn length; SL, seed length; SW, seed width; SPI, spikelets per inflorescence; FLS, florets per spikelet; SS (%), seed set percentage; SI, seeds per inflorescence; TSW, thousand-seed weight; SYP, seed yield per plant. ns: not significant. *, ** and ***: significant at 0.05, 0.01, 0.001 probability level.

**Table 3 plants-08-00561-t003:** Means and results of the ANOVA of awn length, seed yield components and seed yield per plant of 20 Siberian wildrye populations.

Populations	AL (mm)	SL (mm)	SW (mm)	SPI	FLS	SS (%)	SI	TSW (g)	SYP (g)
XH01	12.05ef	9.73a–e	1.65a–e	22.69e–h	6.47a	58.19b–e	86.44ab	4.00a–e	8.96bc
XH02	12.51ef	9.78a–d	1.70a–d	25.73a–d	6.16a	58.07c–e	92.28a	4.01a–d	9.50b
XH03	13.33c–e	10.41ab	1.66a–e	20.64hi	6.34a	53.36e	72.56c	4.03a–e	8.25fg
XH04	9.43h	8.61fg	1.57b–e	21.45g–i	6.02a	72.74ab	90.00a	3.32ef	7.96c–f
XH05	10.78f–h	8.97d–g	1.56b–e	22.25f–h	5.88a	71.24a–c	92.71a	3.51c–f	8.67b–e
LQ01	13.00ef	8.37g	1.59a–e	26.83a–c	6.08a	56.65de	91.83a	3.91b–e	8.80gh
LQ02	12.49ef	9.00d–g	1.54de	21.70g–i	6.14a	55.06de	84.80ab	3.82b–e	7.88bc
LQ03	14.20cd	8.47fg	1.65a–e	27.26ab	6.16a	55.83de	93.19a	4.12a–d	9.13b–d
LQ04	11.98d–g	9.04c–g	1.55c–e	24.74b–f	6.16a	60.87a–e	92.54a	3.83b–e	8.82b–d
LQ05	11.92d–g	9.28c–g	1.61a–e	24.61b–f	5.84a	64.39a–e	92.03a	3.84b–e	8.91h
LQ06	6.30i	8.62fg	1.60a–e	22.48f–h	6.06a	73.09a	92.30a	3.05f	7.60b–d
HZ01	11.37e–h	8.75e–g	1.67a–e	25.02a–f	5.80a	68.34a–d	93.86a	4.21a–c	8.85e–g
TZ01	11.45e–h	9.41b–f	1.51e	23.63d–g	6.12a	63.99a–e	92.14a	3.94b–e	8.27d–g
TZ02	9.76gh	8.87d–g	1.56b–e	22.28f–h	6.16a	66.36a–e	90.14a	3.37d–f	8.38bc
TZ03	11.70e–h	10.07a–c	1.59a–e	24.16c–g	6.10a	63.11a–e	91.87a	3.29ef	9.06bcd
ZN01	15.45bc	10.04a–c	1.76a	27.77a	6.18a	53.94e	77.57bc	4.41ab	8.96bc
ZN02	19.39a	10.48a	1.72a–c	19.34i	6.12a	38.50f	55.25d	4.70a	9.26b
ZN03	17.20ab	10.41ab	1.73ab	24.53b–f	5.96a	56.30de	68.09c	4.57ab	9.99a
ZN04	11.62e–h	9.79a–d	1.56b–e	23.99d–g	6.14a	63.18a–e	92.76a	3.83b–e	8.99bc
ZN05	12.26ef	9.82a–d	1.69a–e	25.31a–e	6.02a	60.90a–e	92.92a	3.96a–e	9.22b

AL, awn length; SL, seed length; SW, seed width; SPI, spikelets per inflorescence; FLS, florets per spikelet; SS (%), seed set percentage; SI, seeds per inflorescence; TSW, thousand-seed weight; SYP, seed yield per plant. Means followed by different letters within the same column are significantly different at *p* < 0.05 probability level.

**Table 4 plants-08-00561-t004:** Correlation coefficients among location, seed yield components and seed yield per plant among 20 Siberian wildrye populations.

Traits	Lat	Long	Alt (m)	SL (mm)	SW (mm)	SPI	FLS	SS (%)	SI	TSW (g)	SYP (g)	AL (mm)
SL (mm)	0.047 ^ns^	−0.663 **	−0.344 ^ns^	1.000								
SW (mm)	−0.196 ^ns^	−0.331 ^ns^	0.074 ^ns^	0.428 **	1.000							
SPI	−0.056 ^ns^	−0.127 ^ns^	0.087 ^ns^	−0.106 ^ns^	0.111 ^ns^	1.000						
FLS	0.196 ^ns^	−0.135 ^ns^	−0.364 ^ns^	0.144 *	0.132 ^ns^	−0.011 ^ns^	1.000					
SS (%)	0.121 ^ns^	0.299 ^ns^	−0.061 ^ns^	−0.369 **	−0.206 **	0.011 ^ns^	−0.473 **	1.000				
SI	0.167 ^ns^	0.403 ^ns^	−0.076 ^ns^	−0.447 **	−0.278 **	0.312 **	−0.062 ^ns^	0.519 **	1.000			
TSW (g)	−0.182 ^ns^	−0.385 ^ns^	0.041 ^ns^	0.466 **	0.604 **	0.094 ^ns^	0.059 ^ns^	−0.418 **	−0.493 **	1.000		
SYP (g)	−0.128 ^ns^	−0.487 *	−0.161 ^ns^	0.487 **	0.448 **	0.371 **	−0.025 ^ns^	−0.231 **	−0.117 ^ns^	0.526 **	1.000	
AL (mm)	−0.189 ^ns^	−0.400 ^ns^	0.094 ^ns^	0.474 **	0.345 **	0.069 ^ns^	0.054 ^ns^	−0.677 **	−0.722 **	0.721 **	0.544 **	1.000

SL, seed length; SW, seed width; SPI, spikelets per inflorescence; FLS, florets per spikelet; SS (%), seed set percentage; SI, seeds per inflorescence; TSW, thousand-seed weight; SYP, seed yield per plant; AL, awn length; Lat, latitude; Long, longitude; Alt, altitude. ^ns^ Non-significant; * *p* < 0.05; ** *p* < 0.01.

**Table 5 plants-08-00561-t005:** The first three principal components of awn length, yield components and seed yield per plant by analyses across 20 populations of Siberian wildrye.

Eigenanalysis	Characteristic	Principal components		
		PC1	PC2	PC3
Variation explained	Eigenvalue	3.814	1.554	1.170
	Contribution rate (%)	42.379	17.270	12.999
	Cumulative rate (%)	42.379	59.648	72.648
Eigenvectors	Traits			
	AL (mm)	0.886	−0.059	−0.099
	TSW (g)	0.832	0.169	−0.134
	SL	0.706	−0.031	−0.113
	SI	−0.702	0.491	0.295
	SS (%)	−0.696	0.364	−0418
	SYP (g)	0.641	0.593	−0.002
	SW (mm)	0.630	0.298	−0.025
	SPI	0.051	0.773	0.399
	FLS	0.227	−0.331	0.841

AL, awn length; TSW, thousand-seed weight; SL, seed length; SI, seeds per inflorescence; SS (%), seed set percentage; SYP, seed yield per plant; SW, seed width; SPI, spikelets per inflorescence; FLS, florets per spikelet.

**Table 6 plants-08-00561-t006:** Means and results of the ANOVA of seed dispersal distance and germination indices of 20 Siberian wildrye populations.

Populations	SDD (m)	GP (%)	GI	GDU (days)	GDE (days)
XH01	1.33a–c	90.00ab	2.98ab	3.80a	3.00c
XH02	1.36a–c	92.00a	3.02ab	4.00a	3.00c
XH03	1.23bc	85.00ab	2.75a–d	4.00a	3.30a–c
XH04	1.06c	95.00a	2.71a–d	4.30a	3.00c
XH05	1.34a–c	96.00a	2.48b–d	4.80a	3.10bc
LQ01	1.23bc	96.00a	2.46b–d	4.70a	3.40a–c
LQ02	1.28bc	96.00a	2.49b–d	4.90a	3.20bc
LQ03	1.47ab	94.00a	2.34cd	4.40a	3.30a–c
LQ04	1.23bc	90.00ab	2.84a–c	4.20a	3.00c
LQ05	1.24bc	86.00ab	2.47b–d	5.10a	3.20bc
LQ06	1.23bc	97.00a	2.24d	4.20a	3.80a
HZ01	1.68a	95.00a	2.21d	5.00a	3.60ab
TZ01	1.30a–c	90.00ab	3.10a	4.20a	3.00c
TZ02	1.07c	95.00a	2.87a–c	4.00a	3.00c
TZ03	1.40a–c	93.00a	3.01ab	3.80a	3.20bc
ZN01	1.35a–c	93.00a	2.50b–d	5.20a	3.20bc
ZN02	1.02c	78.00b	2.36cd	5.00a	3.30a–c
ZN03	1.51ab	87.00ab	2.48b–d	4.40a	3.00c
ZN04	1.20bc	86.00ab	2.57a–d	5.00a	3.10bc
ZN05	1.24bc	95.00a	2.95ab	4.70a	3.00c

SDD, seed dispersal distance; GP, germination percentage; GI, germination index; GDU, germination duration; GDE, germination delay. Means followed by different letters within the same column are significantly different at *p* < 0.05 probability level.

**Table 7 plants-08-00561-t007:** Results of correlation analyses between seed characteristics, germination indices and seed dispersal distance.

Seed Traits	GP (%)	GI	GDU (days)	GDE (days)	SDD (m)
AL (mm)	−0.357 **	−0.080 ^ns^	0.075 ^ns^	−0.057 ^ns^	0.059 ^ns^
SL (mm)	−0.313 **	0.132 ^ns^	−0.059 ^ns^	−0.105 ^ns^	0.034 ^ns^
SW (mm)	−0.140 *	−0.241 **	0.050 ^ns^	0.139 *	0.121 ^ns^
TSW (g)	−0.241 **	−0.084 ^ns^	0.088 ^ns^	0.031 ^ns^	0.108 ^ns^

AL, awn length, SL, seed length; SW, seed width; TSW, thousand-seed weight; GP, germination percentage; GI, germination index; GDU, germination duration; GDE, germination delay; SDD, seed dispersal distance. ^ns^ Non-significant; * *p* < 0.05; ** *p* < 0.01.

**Table 8 plants-08-00561-t008:** The first three principal components of seed traits, germination indices and seed dispersal distance.

Eigenanalysis	Characteristic	Principal Components		
		PC1	PC2	PC3
Variation explained	Eigenvalue	2.762	1.69	1.102
	Contribution rate (%)	30.685	18.78	12.249
	Cumulative rate (%)	30.685	49.466	61.714
Eigenvectors	Traits			
	TSW (g)	0.858	0.118	0.110
	AL (mm)	0.803	0.174	−0.132
	SW (mm)	0.717	−0.069	0.327
	SL (mm)	0.682	0.396	−0.043
	GP (%)	−0.516	0.136	0.479
	GI	−0.276	0.821	0.059
	GDE (days)	0.072	−0.644	0.358
	GDU (days)	0.168	−0.605	−0.390
	SDD (m)	0.161	−0.104	0.672

TSW, thousand-seed weight; AL, awn length; SW, seed width; SL, seed length; GP, germination percentage; GI, germination index; GDU, germination duration; GDE, germination delay; SDD, seed dispersal distance.

**Table 9 plants-08-00561-t009:** Wildrye populations used in the study and their geographic information.

Code	Populations	Location	Latitude (N)	Longitude (E)	Altitude (m)
P1	XH01	Xiahe, Gansu, China	35°13.267′	102°49.067′	2550
P2	XH02	Xiahe, Gansu, China	35°13.267′	102°49.067′	2550
P3	XH03	Xiahe, Gansu, China	35°12.392′	102°40.479′	2770
P4	XH04	Xiahe, Gansu, China	35°11.270′	102°30.313′	2960
P5	XH05	Xiahe, Gansu, China	35°04.138′	102°22.895′	3140
P6	LQ01	Lintan, Gansu, China	35°06.351′	102°24.655′	3080
P7	LQ02	Lintan, Gansu, China	34°35.167′	102°29.535′	3110
P8	LQ03	Lintan, Gansu, China	34°33.793′	102°34.437′	3080
P9	LQ04	Lintan, Gansu, China	34°32.108′	102°37.887′	3050
P10	LQ05	Lintan, Gansu, China	34°32.108′	102°37.887′	3050
P11	LQ06	Lintan, Gansu, China	34°29.782′	102°40.725′	3010
P12	HZ01	Hezuo, Gansu, China	34°05.662′	102°37.927′	3380
P13	TZ01	Tianzhu, Gansu, China	36°57.456′	103°07.778′	2370
P14	TZ02	Tianzhu, Gansu, China	36°57.703′	103°08.896′	2430
P15	TZ03	Tianzhu, Gansu, China	34°56.489′	102°54.985′	2960
P16	ZN01	Zhuoni, Gansu, China	34°50.216′	103°09.503′	3200
P17	ZN02	Zhuoni, Gansu, China	34°41.015′	103°14.835′	3160
P18	ZN03	Zhuoni, Gansu, China	34°34.037′	103°31.393′	2530
P19	ZN04	Zhuoni, Gansu, China	34°25.071′	103°35.153′	2540
P20	ZN05	Zhuoni, Gansu, China	36°57.357′	103°08.708′	2380

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
