# Peer review of "Potential Effects of Awn Length Variation on Seed Yield and Components, Seed Dispersal and Germination Performance in Siberian Wildrye (Elymus sibiricus L.)"

_plants, 2019, doi:10.3390/plants8120561_

Round 1

Reviewer 1 Report

Dear Authors,

This is an interesting manuscript. My specific comments are to be found in the attached file.

I am asking for some specific descriptions of material and Methods. The main improvements of the manuscript will be on the writing The should several times emphasise on main and minor relationships demonstrated by correaltion analyses. Shortening of the text will be possible by fewer citations of figures from tables. This  might ease the communication to the readers of the outcome of the experiment.

In the chapter of Discussion I should have liked to see some more on possible physiological mechanisms which have led to the various relationships. Which kind of adapations may have led to the positive and significant correlation between seed yield per plant and longitude of Place of origin?

specific comment in the attached document.

Author Response

                                                Dr. Wengang Xie

                                                Department of Plant genetics and breeding

                                                Lanzhou University

                                                Lanzhou, 730020 , China

                                                 November 18, 2019

Subject: Submission of Revised Manuscript ID plants-635125

Dear Reviewer,

We thank you for the time and efforts you have put into assessing the previous version of our manuscript.

Title: Potential Effects of Awn Length Variation on Seed Yield and Components, Seed Dispersal, and Germination Performance in Siberian wildrye (Elymus sibiricus L.)

Authors: Fabrice Ntakirutimana, Bowen Xiao, Wengang Xie*, Junchao Zhang, Zongyu Zhang, Na Wang and Jiajun Yan

Your comments and suggestions are all valuable and very helpful for revising and improving our manuscript, as well as the important guiding significance to our researches. After studying the comments and suggestions, we have carefully revised our manuscript. Now, we are pleased to submit the revised version to you. In the response, all the comments and suggestions have been answered in a point-by-point fashion. The major changes we have made to the original manuscript are highlighted in colored text.

Thank you again for your attention to our manuscript!

Sincerely yours,

Dr. Wengang Xie

Point-by-point responses to the comments

Comments and suggestions to the Author: This is an interesting manuscript. My specific comments are to be found in the attached file.

Response: Thank you very much for your attention! We appreciate so much for the warm and encouraging comments from you! According to your suggestions and comments, we have revised and rewritten our manuscript thoroughly.

I am asking for some specific descriptions of material and Methods. The main improvements of the manuscript will be on the writing. They should several times emphasize on main and minor relationships demonstrated by correlation analyses. Shortening of the text will be possible by fewer citations of figures from tables. This might ease the communication to the readers of the outcome of the experiment.

Response: Thank you very much for your attention and kind suggestions! We appreciate so much for the comments from you! According to your suggestions, we have revised and rewritten the mentioned part of the manuscript. We have also revised the manuscript and corrected all the writing mistakes you mentioned. We have also summarized the results as you mentioned (Please check lines 93-263). Further details are found in the responses of specific comments below.

In the chapter of Discussion I should have liked to see some more on possible physiological mechanisms which have led to the various relationships. Which kind of adapations may have led to the positive and significant correlation between seed yield per plant and longitude of Place of origin?

Response: Thank you very much for your attention and kind suggestions! We appreciate so much for the comments from you! We agree with you that emphasizing physiological mechanisms underlying the relationships between awn length and yield components should be well discussed. Thus, we have used previous analyses and added this information. Please check lines 279 and 287.

The distribution of populations varied greatly and this caused variations in the relationships between seed traits and geographical factors. The study didn’t access the data of specific climatic conditions (rainfall, wind, temperature, etc...) of each area. However, we have used previous studies to discuss possible geographical conditions related to these relationships. Please check lines 341-345)

Specific Comments:
Comment 1: line 41: ... genotype has been ...

Response: Thanks a lot for your attention and kind suggestion! We have revised and correct the sentence accordingly. Please check line 40.
Comment 2: I had expected to meet the chapter Materials and Methods her before the chapter of Results.  M & M is now found from line 357 through line 429. I missed during reading chapter Results.

Response: Thanks a lot for the comment and helpful suggestion! We agree with you that the part of materials and methods usually comes before the results part, but according to the structure of this journal (plants), materials and methods’ part comes after results and discussion parts.

Comment 3: Adding information to the tile of table 5
Response: Thanks a lot for the comment! We added this information according to your suggestions (Please check the title of table 5, lines 185-186).
Comment 4: Do not refer all the figures from the table. The figures are found therein. Instead give a statement on what are the main effects, and which ones are the minor ones.

Response: Thank you very much for your kind suggestion. We have carefully revised the statement and rewritten it according to your suggestions. (Please check line 96-98).

Comments 5: writing mistakes for line 92,103
Response: Thank you very much for your attention and kind suggestion! We have corrected these mistakes and written the statement basing on your suggestion (Please check lines 99,137).
Comment 6: Make a new paragraph starting with: Seed lengths ranged.....

Response: Thank you very much for your attention and kind suggestion! Based on your suggestion we have made this paragraph (Please check line 124-135). 
Comment 7: Writing mistakes in table 3 (Please check line 123).

Response: Thank you very much for your attention and kind suggestion! Based on your suggestion, we have corrected these mistakes (Please check table 3, line 146). 

Comment 8: Drop here. The information is given in chapter Materials and Methods. 

Response: Thank you very much for your attention and kind suggestion! We have dropped the statement (Please check line 152).

Comment 9: writing mistakes from lines 136, 137, and 140

Response: Thank you very much for your attention and kind suggestion! We have corrected the mistakes and carefully written the mentioned statements (Please check line 146, 147, 149-150).

Comment 10: Correction of mistakes from table 4 
Response: Thank you very much for your attention and kind suggestion! We have corrected the table and all mistakes were cleared out (Please table 3, line 164-165).

Comment 11:  writing mistakes from lines 150,152,155,157, 160, and 167 
Response: Thank you very much for your attention and kind suggestion! We have corrected mistakes accordingly (Please check lines 154, 156-157,162,167, and 185-186).

Comment 12: Writing mistakes from lines 176,181,184,185,186-189,191-192,195,198, and 203.

Response: Thank you very much for your attention and kind suggestion! We agree that the statements mentioned in these lines some had grammatical errors others missed useful information. Thus, we corrected these statements based on your suggestions; please check lines 191,197, 199,201-205, 208, 215.

Comment 13: This information belongs to the chapter Materials and Methods. Drop it here.

Response: Thank you very much for your attention and kind suggestion! We dropped the unnecessary information as you suggested (please check lines 229).

Comment 14: Express also here 'geographical factors' as geographical variation in place of origin (collection). Because of my reading of Results before Materials and Methods I did not know whether these factors were

Response: Thank you very much for your attention and kind suggestion! We have referred to your suggestions and added the information mentioned (please check lines 340-345).

Comment 15: What is 'dispersal unit'?

Response: Thank you very much for your attention and kind suggestion! We have referred to your comment and provided clear information (please check lines 367-268).
Comment 16: What is 'dispersal unit'?

Response: Thank you very much for your attention and kind suggestion! We have referred to your comment and provided a clear information (please check lines 353).

Comment 17:  writing mistakes from lines 349,355,374,385-386
Response: Thank you very much for your attention and kind suggestion! We have corrected mistakes accordingly (Please check lines 364, 392,398-400, 426,439, and 440).

Comment 18: Wind speed? Define the wind speed in m per second.

Response: Thank you very much for your attention and kind suggestion! We have mentioned the wind speed of the fan used and added some information related to the measurement (Please check lines 438-440).

Comment 19: Correcting typing error; probably an 'electric' fan, not 'electronic'?

Response: Thank you very much for your attention and kind suggestion! We have corrected mistakes accordingly (Please check lines 443,447).

Comment 19: Does that mean that the 'seminal root' was 1 mm long (and visible)??

Response: Thank you very much for your attention and kind suggestion! We have provided clear information according to your suggestions (Please check lines 558-459).

Comment 20: Writing mistakes from lines 407 and 412

Response: Thank you very much for your attention and kind suggestion! We have corrected mistakes accordingly (Please check lines 463,469).

Comment 21: Explain 'n=200'. How has this number originated from the 20 populations?

Response: Thank you very much for your attention and comment! We this number originated from 10 plants of each of 20 Siberian wildrye populations. We have added this information as you mentioned (Please check lines 474-476).

Comment 22: Writing mistakes from lines 435,436,437,439, and 442-443?

Response: Thank you very much for your attention and kind suggestion! We have corrected these mistakes basing on your suggestions (Please check lines 484,486,488, and 491-492).

Thank you again for your attention to our manuscript!

Reviewer 2 Report

The article discusses the importance of awns length and its impact on agronomic performance and ecophysiological characteristics. The work was done on sufficient collection accessions of wildrye.

The manuscript is very interesting by the scientific approach and the subject of agronomic interest. However, there is a difficulty in reconciling the ecophysiological and agronomic characteristics.

In addition, the main questioning in this manuscript and which greatly limits its scope is the used statistical approach.

Why did the authors choose to use Path correlation analysis?

Now the question is to see the importance of the weight of the variables in the realization of the yield and the germination, the dispersion of the seeds. Why not choose the step-by-step regression that makes this assessment of the weight of each variable on the main factor?

Or

Choose multifactor analysis as the principal component analysis. The latter makes it possible to add additional parameters (such as environmental parameters) without influencing the correlations of the main factors studied.

Or to do analysis of variance including covariates such as environmental factors.

Other remarks

The effects of awns of cereals have been extensively studied. More recent studies deserve to be added to this work. For example the works done by the team of Philippe Monneveux in France or José Luis Araus in Spain.

Participation of Green Organs to Grain Filling in Triticum turgidum durum var Grown under Mediterranean Conditions. International Journal of Molecular Sciences 19 (1), 56.

Contribution of Different Organs to Grain Filling in Durum Wheat under Mediterranean Conditions I. Contribution of Post-Anthesis Photosynthesis and Remobilization. Journal of Agronomy and Crop Science 201 (5), 344-352.

Author Response

                                                Dr. Wengang Xie

                                                Department of Plant genetics and breeding

                                                Lanzhou University

                                                Lanzhou, 730020 , China

                                                 November 18, 2019

Subject: Submission of Revised Manuscript ID plants-635125

Dear Reviewer,

We thank you for the time and efforts you have put into assessing the previous version of our manuscript.

Title: Potential Effects of Awn Length Variation on Seed Yield and Components, Seed Dispersal, and Germination Performance in Siberian wildrye (Elymus sibiricus L.)

Authors: Fabrice Ntakirutimana, Bowen Xiao, Wengang Xie*, Junchao Zhang, Zongyu Zhang, Na Wang and Jiajun Yan

Your comments and suggestions are all valuable and very helpful for revising and improving our manuscript, as well as the important guiding significance to our researches. After studying the comments and suggestions, we have carefully revised our manuscript. Now, we are pleased to submit the revised version to you. In the response, all the comments and suggestions have been answered in a point-by-point fashion. The major changes we have made to the original manuscript are highlighted in colored text.

Thank you again for your attention to our manuscript!

Sincerely yours,

Dr. Wengang Xie

Point-by-point responses to the comments

Comments and suggestions to the Author:

Comment 1.The article discusses the importance of awns length and its impact on agronomic performance and ecophysiological characteristics. The work was done on sufficient collection accessions of wildrye.

Comment 2.The manuscript is very interesting by the scientific approach and the subject of agronomic interest. However, there is a difficulty in reconciling the ecophysiological and agronomic characteristics.

Response for comment 1&2: Thank you very much for your attention! We appreciate so much for the warm and encouraging comments from you! According to your suggestions, we have revised and rewritten our manuscript thoroughly.

Comment 3. In addition, the main questioning in this manuscript and which greatly limits its scope is the used statistical approach.

Why did the authors choose to use Path correlation analysis?

Now the question is to see the importance of the weight of the variables in the realization of the yield and the germination, the dispersion of the seeds. Why not choose the step-by-step regression that makes this assessment of the weight of each variable on the main factor?

Or

Choose multifactor analysis as the principal component analysis. The latter makes it possible to add additional parameters (such as environmental parameters) without influencing the correlations of the main factors studied.

Or to do analysis of variance including covariates such as environmental factors.

Response: Thank you very much for your attention and kind suggestions! We appreciate so much for the comments from you! All the analyses you mentioned are very important for the improvement of our manuscript. Thus, based on several statistical analyses you suggested, we have chosen principal component analysis (PCA) as it may summarize the variables in an easy-to-read format. This analysis replaced the path correlation analysis used in the previous version of this manuscript. Please check lines 174-183, 245-258. This information on PCA is also found in Tables 5 and 6. These analyses were also included in the part of materials and methods (Please check lines 472, 476-478)   

Other remarks

The effects of awns of cereals have been extensively studied. More recent studies deserve to be added to this work. For example the works done by the team of Philippe Monneveux in France or José Luis Araus in Spain.

Participation of Green Organs to Grain Filling in Triticum turgidum durum var Grown under Mediterranean Conditions. International Journal of Molecular Sciences 19 (1), 56.

Contribution of Different Organs to Grain Filling in Durum Wheat under Mediterranean Conditions I. Contribution of Post-Anthesis Photosynthesis and Remobilization. Journal of Agronomy and Crop Science 201 (5), 344-352.

Response to these remarks

Thank you very much for your attention and kind suggestions! We appreciate so much for the comments from you! All the previous works you mentioned are very important for the improvement of our manuscript and are deserved to be included in the manuscript. Thus, we have read all these manuscripts and improved our manuscript using the information from these works. Please check lines 58-64, 272-278.

Thank you again for your attention to our manuscript!

Reviewer 3 Report

This manuscript studies Siberian wildrye awn length and yield performance. Awn length was already studied in many grass species, dealing with grain yield, productivity, and natural habitat adaptation. The manuscript is within the scope of Plants and explores new information about these characteristics in Siberian wildrye. But, the manuscript suffers from many problems in data analysis, conclusions and writing. Overall, in my opinion, this manuscript is not adequate for publication as it is now. The manuscript needs major revision. The authors must provide acceptable response to the requested critical missing information and revise the manuscript accordingly.

Major comments:

1- Most of the manuscript deals with yield and yield components, and many parameters have been determined (tables 2,3,4…). However, it is not clear how the authors got the GYP. I tried to calculate it by other yield components and could not reach the same (or similar) grain yield (table 3). Therefore, interpretation is problematic. Authors must make this point clear and add yield equation.     

2– It is very hard to read the paper. Writing must be concise and should focus on the main results. Please proofread it prior to submission. By that, figures 1 and 2 will be in the right place and with the correct title… the Materials and Methods should appear after Conclusion, etc. In addition, I found many mistakes and unclear statements, for example: Table 2- FLP or FLS; line 125- left side!; What is “seed set”;  line 51- “believed”? line 71- emerge from a deeper layer is affected by coleoptile!; line 94- I see more than two groups; line 245- authors did not show that the awns reduced the seed set, they maybe exhibit a negative correlation.

3- Statistics: 1- Duncan test is not acceptable, use LSD or LSMeans Differences Tukey HSD test. 2- Seeds were collected from different basal/central/distal spikelet, why authors did not test 2-way ANOVA? 3- Seed dispersal distance is affected mainly by seed weight, therefore analysis of awn effect must be covariate with seed weight. 4- PCA term is problematic, as it usually stands for Principal Components Analysis. 5- The Path analysis did not contribute much more information, I think that MRA and PCA would be better. 6- Lat/Long/Alt factors represent much more than just the location. At least the location determines the day length, temperature and precipitation that prevail in that location during the growing season. All these environment parameters must be added and analyzed as they have a direct effect on plant growth and yield production.  

4- Authors must increase the visibility of the population sites that are included in Fig 3. Reader should be able to connect data from other tables to the location in fig 3.

5– The discussion must include a comprehensive clear comparison to all known information from other species having awn length variation (from null to very long. For wheat these data must be related to durum and aestivum wheat that varied in awn formation. Moreover, as awn makes photosynthesis, authors must clarify the source-sink issue. I do not agree that awn competes with grain filling…

Author Response

                                                Dr. Wengang Xie

                                                Department of Plant genetics and breeding

                                                Lanzhou University

                                                Lanzhou, 730020 , China

                                                November 18, 2019

Subject: Submission of Revised Manuscript ID plants-635125

Dear Reviewer,

We thank you for the time and efforts you have put into assessing the previous version of our manuscript.

Title: Potential Effects of Awn Length Variation on Seed Yield and Components, Seed Dispersal, and Germination Performance in Siberian wildrye (Elymus sibiricus L.)

Authors: Fabrice Ntakirutimana, Bowen Xiao, Wengang Xie*, Junchao Zhang, Zongyu Zhang, Na Wang and Jiajun Yan

Your comments and suggestions are all valuable and very helpful for revising and improving our manuscript, as well as the important guiding significance to our researches. After studying the comments and suggestions, we have carefully revised our manuscript. Now, we are pleased to submit the revised version to you. In the response, all the comments and suggestions have been answered in a point-by-point fashion. The major changes we have made to the original manuscript are highlighted in colored text.

Thank you again for your attention to our manuscript!

Sincerely yours,

Dr. Wengang Xie

Point-by-point responses to the comments

Comments and suggestions to the Author:

This manuscript studies Siberian wildrye awn length and yield performance. Awn length was already studied in many grass species, dealing with grain yield, productivity, and natural habitat adaptation. The manuscript is within the scope of Plants and explores new information about these characteristics in Siberian wildrye. But, the manuscript suffers from many problems in data analysis, conclusions and writing. Overall, in my opinion, this manuscript is not adequate for publication as it is now. The manuscript needs major revision. The authors must provide acceptable response to the requested critical missing information and revise the manuscript accordingly.

Response: Thank you very much for your attention! We appreciate so much for constructing comments from you! According to your suggestions, we have revised and rewritten our manuscript thoroughly.

Major comments:

1- Most of the manuscript deals with yield and yield components, and many parameters have been determined (tables 2,3,4…). However, it is not clear how the authors got the GYP. I tried to calculate it by other yield components and could not reach the same (or similar) grain yield (table 3). Therefore, interpretation is problematic. Authors must make this point clear and add yield equation.

Response. Thank you very much for your attention and kind suggestions! We agree with you that the earlier version of the manuscript didn’t demonstrate clearly how we obtained SYP. SYP (g plant-1) was measured at harvesting maturity after the seeds of each collected plant were threshed, and cleaned. We didn’t use the yield equation to determine seed yield. Therefore, we added further information to better clarify this measurement. Please check lines 421-428.

2– It is very hard to read the paper. Writing must be concise and should focus on the main results. Please proofread it prior to submission. By that, figures 1 and 2 will be in the right place and with the correct title… the Materials and Methods should appear after Conclusion, etc. In addition, I found many mistakes and unclear statements, for example: Table 2- FLP or FLS; line 125- left side!; What is “seed set”;  line 51- “believed”? line 71- emerge from a deeper layer is affected by coleoptile!; line 94- I see more than two groups; line 245- authors did not show that the awns reduced the seed set, they maybe exhibit a negative correlation.

Response: Thank you very much for your attention and kind suggestions! We appreciate so much for the comments from you! We have summarized the results part, as you mentioned, and provided clear and concise information. Please check lines 92-263, therein you can found highlighted changes.

For the mistakes you mentioned from lines 51, 71, 94, and 245, we have corrected them accordingly. Please check lines 50, 78-80, 99-101. We also added information to better clarify the heat map (fig 1), please check lines 100-105. We added information about the relationship between seed set(%) and awn length (Please check lines 270-272).

3- Statistics: 1- Duncan test is not acceptable, use LSD or LSMeans Differences Tukey HSD test.

Response: Thank you very much for your attention and kind suggestions! As you suggested, we have re-analyzed the data and used Tukey HSD to compare means differences. Please check lines Tables 3 and 6. We mentioned this statistical test also in the lines 464-466.

2- Seeds were collected from different basal/central/distal spikelet, why authors did not test 2-way ANOVA? Thank you very much for your attention and kind suggestions! We agree with you that 2-way ANOVA could add useful information to the manuscript. Thus, we conducted it and provided a summarytable (Table 2). We also reported some major results of this analysis, Please check lines 116-119.

3- Seed dispersal distance is affected mainly by seed weight, therefore analysis of awn effect must be covariate with seed weight. 4- PCA term is problematic, as it usually stands for Principal Components Analysis. 5- The Path analysis did not contribute much more information, I think that MRA and PCA would be better. 6- Lat/Long/Alt factors represent much more than just the location. At least the location determines the day length, temperature and precipitation that prevail in that location during the growing season. All these environment parameters must be added and analyzed as they have a direct effect on plant growth and yield production.

Response: Thank you very much for your attention and kind suggestions! We appreciate so much for the comments from you! All the analyses you mentioned are very important for the improvement of the manuscript. Thus, we have used principal component analysis (PCA), as you suggested. This analysis replaced the path correlation analysis used in the previous version of this manuscript. Please check lines 174-183, 245-258. This information on PCA is also found in Tables 5 and 6. The information concerning these analyses was also included in the part of materials and methods (Please check lines 471, 475-477).

For other environmental parameters, we agree with you that this information is very important. However, because the study was conducted in different areas, some of this information was not easy to take and available public meteorological data is not very specific for the sites where we collected the data. For not mislead the readers of the manuscript, we only used the data (Lat/Long/Alt) that we are sure that is more accurate. We really thank you again for this insightful comment, and we will consider it for our future researches.

4- Authors must increase the visibility of the population sites that are included in Fig 3. Reader should be able to connect data from other tables to the location in fig 3.

Response: Thank you very much for your attention and kind suggestions! We appreciate so much for the comments from you! We agree with you that the previous version of figure 3 is not clear, and it is not easy to associate its information with other figures and tables. Thus, we have added information you mentioned for figure 3 and this time it has more visibility of the population sites. Additionally, we added information in table 9 to more clarify the connection between populations and sites of collection presented in figure 3. Please check lines 403-418.

5– The discussion must include a comprehensive clear comparison to all known information from other species having awn length variation (from null to very long. For wheat these data must be related to durum and aestivum wheat that varied in awn formation. Moreover, as awn makes photosynthesis, authors must clarify the source-sink issue. I do not agree that awn competes with grain filling

Response: Thank you very much for your attention and kind suggestions! As you mentioned that the earlier version of the manuscript lacked further information from previous data, we have added information some information (Please check lines 279-287). For the information regarding the source-sink issue, Please check lines 58-64. For the information regarding the completion of long awns and grain filling, we have rewritten the statement and provided more clear information (Please check lines 310-312)

Thank you again for your attention to our manuscript!

Round 2

Reviewer 2 Report

Manuscript was improved by including and performing modifiications. This modifications changed  some parts of this study and results were presented clearly. Statistcal analyses were released as requested and allowed to some modifications which helped authors to present their findings better.

Author Response

                               Dr. Wengang Xie

                               Department of Plant genetics and breeding

                               Lanzhou University

                               Lanzhou, 730020 , China

                                Tel: +86-931 891-3014

                                November 23, 2019

Subject: Submission of Revised Manuscript ID plants-635125

Dear Reviewer,

We thank you for the time and efforts you have provided into assessing the previous versions of our manuscript.

Title: Potential Effects of Awn Length Variation on Seed Yield and Components, Seed Dispersal, and Germination Performance in Siberian wildrye (Elymus sibiricus L.)

Authors: Fabrice Ntakirutimana, Bowen Xiao, Wengang Xie*, Junchao Zhang, Zongyu Zhang, Na Wang and Jiajun Yan

Your comments and suggestions are all valuable and very helpful for revising and improving our manuscript, as well as the important guiding significance to our researches.

We look forward to respond to any further questions and comments you may have.

Thank you again for your attention to our manuscript!

Sincerely yours,

Dr. Wengang Xie

Reviewer 3 Report

The paper is now much better, well written and the author follows a correct methodology, I recommend the paper for publication after minor corrections and clarifications.

The authors forget my previous point: Statistics: 3- Seed dispersal distance is affected mainly by seed weight, therefore analysis of awn effect must be covariate with seed weight.

Author Response

                               Dr. Wengang Xie

                               Department of Plant genetics and breeding

                               Lanzhou University

                               Lanzhou, 730020 , China

                               Tel: +86-931 891-3014

                               November 23, 2019

Subject: Submission of Revised Manuscript ID plants-635125

 Dear Reviewer,

We thank you for the time and efforts you have provided into assessing the previous versions of our manuscript.

Title: Potential Effects of Awn Length Variation on Seed Yield and Components, Seed Dispersal, and Germination Performance in Siberian wildrye (Elymus sibiricus L.)

Authors: Fabrice Ntakirutimana, Bowen Xiao, Wengang Xie*, Junchao Zhang, Zongyu Zhang, Na Wang and Jiajun Yan

Your comments and suggestions are all valuable and very helpful for revising and improving our manuscript, as well as the important guiding significance to our researches. After studying the comments and suggestions you provided for the previous version of the manuscript, we have carefully revised our manuscript. Now, we are pleased to submit the revised version to you. The detailed answers for the comments and suggestions are found below. The major changes we have made to the original manuscript are highlighted in colored text.

Thank you again for your attention to our manuscript!

Sincerely yours,

Dr. Wengang Xie

Responses to the comments

Comments and suggestions to the Author:

The paper is now much better, well written and the author follows a correct methodology, I recommend the paper for publication after minor corrections and clarifications.

Response: Thank you very much for your attention and kind suggestions! We appreciate so much for the warm and encouraging comments from you! According to your suggestions, we have revised and rewritten our manuscript thoroughly.

Specific comment

The authors forget my previous point: Statistics: 3- Seed dispersal distance is affected mainly by seed weight, therefore analysis of awn effect must be covariate with seed weight.

Response

Thank you very much for your attention and kind suggestions! We would like to apologize that we didn’t answer this comment in the previous version of the manuscript. As you mentioned, since seed dispersal is affected mainly by seed weight, thus, awn length must be covariate with seed weight. We agree with you for this statement. To prove it, we have conducted univariate linear regression analyses to test further effects of awn length on seed yield and yield components. Our results clearly showed that TSW (thousand-seed weight) significantly covaried with awn length. We have added this information in the manuscript (Please check Figure 3; lines 191-195). This information was also included in the results part (Please check lines 153-155, 166-167). Besides, the analyses were included in the methodology part (Please check lines 505-506).

Thank you again for your attention to our manuscript!